# Changes in Markers of Oxidative Stress and α-Amylase in Saliva of Children Associated with a Tennis Competition

**DOI:** 10.3390/ijerph17176269

**Published:** 2020-08-28

**Authors:** José María Giménez-Egido, Raquel Hernández-García, Damián Escribano, Silvia Martínez-Subiela, Gema Torres-Luque, Enrique Ortega-Toro, José Joaquín Cerón

**Affiliations:** 1Department of Physical Activity and Sport, Faculty of Sport Science, Regional Campus of International Excellence “Campus Mare Nostrum”, University of Murcia, 30720 Murcia, Spain; josemaria.gimenez@um.es (J.M.G.-E.); eortega@um.es (E.O.-T.); 2Sport Performance Analysis Association, University of Murcia, 30720 Murcia, Spain; gtluque@ujaen.es; 3Interdisciplinary Laboratory of Clinical Analysis of the University of Murcia (Interlab-UMU), Regional Campus of International Excellence ‘Campus Mare Nostrum’, University of Murcia, Campus de Espinardo s/n, 30100 Espinardo, Spain; silviams@um.es (S.M.-S.); jjceron@um.es (J.J.C.); 4Faculty of Humanities and Education Sciences, University of Jaen, 23071 Jaen, Spain

**Keywords:** childhood, redox regulation, exercise

## Abstract

The purpose of this paper was to analyze the changes caused by a one-day tennis tournament in biomarkers of oxidative stress and α-amylase in saliva in children. The sample was 20 male active children with the following characteristics: (a) age of players = 9.46 ± 0.66 years; (b) weight = 34.8 ± 6.5 kg; (c) height = 136.0 ± 7.9 cm; (d) mean weekly training tennis = 2.9 ± 1.0 h. The tennis competition ran for one day, with four matches for each player. Data were taken from the average duration per match and the rating of perceived exertion (RPE). Four biomarkers of antioxidant status: uric acid (AU), Trolox equivalent antioxidant capacity (TEAC), ferric reducing ability of saliva (FRAS, cupric reducing antioxidant capacity (CUPRAC) and salivary alpha-amylase (sAA) as a biomarker of psychological stress were measured in saliva. The time points were baseline (at home before the tournament), pre-competition (immediately before the first match) and post-match (after each match) measurements. The four biomarkers of antioxidant status showed a similar dynamic with lower values at baseline and a progressive increase during the four matches. Overall one-day tennis competition in children showed a tendency to increase antioxidant biomarkers in saliva. In addition, there was an increase in pre-competition sAA possibly associated with psychological stress. Further studies about the possible physiological implications of these findings should be performed in the future.

## 1. Introduction

Tennis is an intermittent sport played of high-intensity actions characterized by short bouts of high intensity exercise, which are composed by accelerations/decelerations, strokes and change-of-direction (COD) interspersed with periods of low-moderate intensity or rest short break between points and moderate rest between games and sets [1]. This leads to repetitive and elevated muscle demands during the match that induce a wide variety of physiological responses, including the increased production of reactive oxygen species (ROS) [2]. Several studies have shown that prolonged endurance exercise or short duration/high intensity exercise results in increased oxidative stress [3,4,5,6]. This increase in oxidative stress can lead to metabolic adaptations involving various biological functions such as maintenance of tissue homeostasis, cell proliferation and differentiation, cell migration and regulation of transcriptional activity [7,8]. In addition, in situations of acute exercise the oxidative stress can produce increases in antioxidant compounds [9,10,11].

Despite its physiological importance, scarce information is available regarding the effects of repeated exercise on oxidative stress in healthy children [12,13]. High levels of oxidative stress markers in children and adolescents have been reported, possibly due to a high flow of oxygen to the working muscles, and/or due to their immature antioxidant defense systems [14]. Indeed, antioxidant reserves may be lower in children, especially those engaged in chronic intensive training programs; suggesting that children, and probably adolescents, may be susceptible to exercise-induced oxidative damage [14]. So, the knowledge of exercise induced oxidative stress responses would be valuable for children and adolescents engaged in systematic strenuous exercise [12]. However, no studies of oxidative stress biomarkers have been found in children during an exercise made in competition, and in tennis in particular.

The use of saliva as a sample has the advantage of an easy and non-painful collection compared to blood, being very suitable in field conditions and in cases where repeated sampling is required [15]. It is also of special interest in case of children because this collection procedure generates a minimum discomfort and anxiety [15,16]. Saliva has various metabolites and enzymes with oxidant and antioxidant properties, and it has been used to investigate possible changes in the oxidative status in different conditions such as oral and systemic diseases and physical efforts [17,18].

Uric acid contributes to approximately the 70% of the total antioxidant capacity of the saliva and increases after different types of resistance exercise [17,18]. It is a part of the total antioxidant capacity (TAC), which can be measured to assess in a global way the antioxidant status of biological samples and evaluate the antioxidant response against the free radicals produced in physiological or pathological conditions. Trolox equivalent antioxidant capacity (TEAC), ferric reducing ability of saliva (FRAS), and cupric reducing antioxidant capacity (CUPRAC) are different assays described to determine TAC of a sample [19]. Increase of FRAS has been described as a possible positive effect of nutritional supplements in exercise performance [20,21], that can be used to compensate the reduction of TAC occurring in physical activity [22].

Salivary α-amylase (sAA) is a biomarker of the stress experienced by athletes in training and competition [16]. The increases in this enzyme could be due to the fact that exercise causes an increase in the activity of the sympathetic system [10,11,23]. However, it should be noted that psychological stress can also trigger the release of sAA [24].

Despite the growing interest in the study of the physiological responses of children in competitive environments, most studies on this subject deal with the most common motor behavior as a function of the type of surface [25], the assessment of technique in young tennis players [26] and the relationship between the performance of young people and future sporting success [27]. This work aims to shed light on the changes in saliva biomarkers of oxidative and psychological stress in a children’s tennis competition. Therefore, the main objective of this study is to analyze the possible changes of a profile of four antioxidant biomarkers (uric acid, TEAC, FRAS, CUPRAC) and one biomarker of psychological stress (sAA) during a tennis competition in which children have to play four matches in one day.

## 2. Materials and Methods

### 2.1. Design and Subjects

A cross-sectional study was designed to explore the change on biomarkers over time in competition [28,29]. The sample was 20 active male children with the following characteristics: (a) age of players = 9.4 ± 0.6 years; (b) mean weekly training tennis = 2.9 ± 1.0 h; (c) tennis experience = 3.6 ± 1.5 years; (d) weight = 34.8 ± 6.5 kg; (e) height = 136.0 ± 7.9 cm; (f) abdominal perimeter = 64.4 ± 7.6 cm; and (g) VO_2_max = 20.9 ± 4.5 mL·kg^−1^·min^−1^. The non-randomization of the sample was for the sample’s specificity (the criteria selection were accessibility and proximity and the same level of competition). This study respected the ethical principles established by the UNESCO Declaration on Bioethics and Human Rights. Prior to investigating, ethical clearance was obtained from the “Ethics Committee of the University of Murcia” (Spain) (ID 1925/2018). Following the Declaration of Helsinki, the players voluntarily participated in the study and their written informed consent was obtained and signed from the parents/guardians of all participants for the development of this study.

The competition format had the following characteristics: (a) each player played four tennis single matches from one set to four games in one day (except in case of illness or injury); (b) each player was assigned a group randomly, which comprised five players; (c) the competition system was “round robin”; (d) rest time among matches was 23.6 ± 6.0 min to avoid cognitive, emotional and physical fatigue [30]. The competition took place in two shifts, the morning shift from 9 a.m. to 1 p.m. and the afternoon shift from 4 p.m. to 7 p.m. In addition, all matches were played according to the current rules of the International Tennis Federation for under-10 tennis players (green stage). For each match, data on duration and Borg Scale of Perceived Exertion (RPE) of 6–20 was collected immediately after the end of each match [31]:duration and RPE average of the first match: 23.6 ± 12.4 min and 11 ± 4;duration and RPE average of the second match: 23.1 ± 12.7 min and 13 ± 4;duration and RPE average of the third match: 20.4 ± 11.9 min and 13 ± 5;duration and RPE average of the fourth match: 16.6 ± 8.9 min and 12 ± 5.

### 2.2. Procedure and Variables

The participants themselves performed sample collection. All participants received detailed information by oral communication and written guide-lines about the aims and experimental protocol, the saliva collection procedure. They were informed about the need to avoid coughing or clearing the throat into the collection tube and were to abstain from brushing teeth or using mouthwash, ingesting any food or chewing gum for 1 h before saliva collection. The participants rinsed their mouth with water five minutes before saliva collection. Then, unstimulated saliva was collected by passive drool in the absence of chewing movements into 10 mL plain tubes . Collection lasted between 2 and 5 min. The volunteers sat in a relaxed position throughout the sampling procedure to avoid any stress. Between 3 to 5 mL of saliva was obtained from each participant, all samples were checked for blood contamination by visual inspection and no reddish samples indicating blood contamination were included in the study. After collection, the saliva samples were send to the laboratory and centrifuged (Universal 320R, Hettich, Tuttlingen, Germany) at 5000× *g* and 4 °C for 5 min, then the supernatant was collected and divided into aliquots, discarding the sediment.

The baseline samples were collected two hours before the tournament and before breakfast; the pre-competitive sample was collected immediately before the first match; and the post-match samples were collected immediately after the end of each match.

The analytical methods used have been adapted in the authors’ laboratory for saliva samples. Their fundamentals, details of the reagents and analytical performance, have been previously described [32]. All assays were performed on an automated biochemistry analyzer (Olympus AU400, Olympus Diagnostica GmbH, Ennis, Ireland) at 37 °C. A single measurement was made in all cases since all analytical methods showed an intra-assay imprecision lower than 15%, which indicated adequate assay repeatability.

### 2.3. Statistical Analysis

Data analysis was divided into two phases: (a) descriptive analysis; and (b) individual growth model. Descriptive analysis was performed to observer the mean values, standard deviation, variance, kurtosis and skewness for each shot. Individual growth model was selected to see changes in saliva parameters over time, modelled by potential growth patterns through multilevel models. The growth patterns used to evaluate which curve best fits the point cloud at different times were the linear (first-order polynomial), quadratic (second-order polynomial), and cubic (third-order polynomial). The maximum likelihood estimation was selected for its use with incomplete data over other techniques and its modeling capability on growth models with fixed and random effects [33]. An autoregressive covariance structure (AR(1)) was chosen on the assumption that variances would be heterogeneous [34]. To assess the overall fit the model, the value log-likelihood (−2*LL*) was tested for significance with *df* equal to the number of parameters estimated. To compare models, it was added polynomials one at a time to assess the change in chi-square (χ^2^_change_). The change in chi-square was calculated by subtracting the log-likelihood of the new polynomial from the value for the old. The change in degree of freedom (*d*ƒ_change_) was estimated by subtracting the number of parameters (old and new model), in which *k* is the number of parameters in the respective model:χ^2^_change_ = (−2*LL*_old_) − (−2*LL*_new_)*df*_change_ = *k*_old_ − *k*_new_(1)

The new polynomial was included in the model, if the critical change in square had a bigger value than critical value of chi squared statistic (3.84, *p* < 0.05 and 6.63, *p* < 0.01) with 1 degree of freedom (the *dƒ*_change_ is equal between adjacent polynomials). The Akaike’s information criterion (AIC) was included to complement the information on model fit but not tested for significance. The alpha value for evaluating whether the growth functions in the model significantly predict the saliva parameters was set less than 0.05 (test of fixed effects) [35]. Finally, to know how much the intercepts and slopes varied over time (random effects), the importance covariance parameters was estimated. These estimated should be treated with caution, because the type of covariance selected modify the random effects in the model. To minimize this situation AR(1) is often assumed in individual growth models. Statistical analyses were performed using the IBM SPSS Statistics 25.0 statistical package (IBM Corp., Armonk, NY, USA), and spreadsheet Jamovi 1.2.2 based on the R graphical user interface.

## 3. Results

The results show descriptive values of saliva parameters and their trends over time. Table 1 shows the descriptive analysis for every saliva parameter analyzed per shot. The different figures indicate the results related to growth models, showing the trends linear, quadratic and cubic over time for each saliva analyte. A bar plot illustrates the mean values and standard deviation while a scatterplot indicate which are the best polynomial that represents the random effects by player over time.

### 3.1. AU

The bar plot shows that the lowest average value was the baseline increasing this value in the pre-competition almost 1 μmol/L (Figure 1a). Figure 1b indicates that the mean trajectory of AU increased at first and it gradually flatted out from shot 4 (post-match 2). The assessment in chi-square change was significant (χ^2^ (1) = 5.51, *p* < 0.05) between the linear (−2*LL* = 1122.47, AIC = 1134.47) and quadratic polynomial (−2*LL* = 1116.96, AIC = 1130.96), but this was not significate (χ^2^ (1) = 2.68, *p* < 0.05) between the quadratic polynomial and cubic polynomial (−2*LL* = 1114.28, AIC = 1130.28). The test of fixed effects and the parameter estimates showed that the linear, B = 0.509, F_1,325.99_ = 8.71, *p* = 0.003, and quadratic, B = −0.056, F_1,309.20_ = 5.57, *p* = 0.019, trends both significantly described the pattern of the data over time; however, the cubic trend, B = 0.029, F_1,308.21_ = 2.69, *p* = 0.120, does not. The positive linear effect indicates a general tendency of the AU to increase in time, and the negative quadratic effect indicates that the AU trend tends to become flat over time. The estimation of covariance parameters showed that the variance of random intercepts was Var(u_0j_) = 1.34 assuming that baseline AU varied significantly across young tennis player (*p* = 0.13). Also, the variance of the tennis players’ slopes not varied significantly Var(u_1j_) = 0.01. Hence, the change in AU over time not varied significantly across people (*p* = 0.268). Finally, the covariance between the slopes and intercepts (−0.52) suggests that as intercepts increased, the slope decreased, however, this was not statistically significant (*p* = 0.76).

### 3.2. FRAS

Figure 2a shows that the highest change in the average value was between baseline and pre competition shot (13 mmoll). Specifically, Figure 2b shows that the mean values FRAS increased gradually over the time. The assessment in chi-square change not was significant between the linear (−2*LL* = −328.41, AIC = −316.410) and quadratic polynomial (−2*LL* = −337,124, AIC = −322.124). The fixed effects test and parameter estimates that the linear, B = 0.023, F_1,20.80_ = 20.46, *p* = 0.000, trend significantly described the pattern of the data over time. The positive linear effect indicates a general tendency of the FRAS to increase over time. The estimation of covariance parameters showed that the variance of random intercepts was Var(u_0j_) = 0.024, assuming that baseline FRAS varied significantly across young tennis player (*p* = 0.025). Moreover, the variance of the tennis players’ slopes not varied significantly Var(u_1j_) = 0.01. Hence, the change in FRAS over time not varied significantly across people (*p* = 0.380). Finally, the covariance between the slopes and intercepts (0.12) suggests that as intercepts decreased, the slope increased, however, this was not statistically significant (*p* = 0.835).

### 3.3. CUPRA

Figure 3a,b show that the average trajectory of CUPRA increased gradually over the time. The assessment in chi-square change not was significant between the linear (−2*LL* = −873.67, AIC = −861.679) and quadratic polynomial (−2*LL* = −874.254, AIC = −860.245). The test of fixed effects and the parameter estimates showed that the linear, B = 0.065, F_1,329.77_ = 10.47, *p* = 0.001, trend significantly described the pattern of the data over time. 

The positive linear effect indicates a general tendency of the CUPRA to increase over time. The estimation of covariance parameters showed that the variance of random intercepts was Var(u_0j_) = 0.024 assuming that baseline CUPRA varied significantly across young tennis player (*p* = 0.025). The variance of the tennis players’ slopes Var(u_1j_) = 0.00 and covariance between intercepts and slopes (0.99) showed redundant covariance parameters.

### 3.4. TEAC

The bar plot showed a progressive increase in the average values. Figure 4a shows that the average trajectory of TEAC increased gradually over the time. The assessment in chi-square change not was significant between the linear (−2*LL* = −693.71, AIC = −681.71) and quadratic polynomial (−2*LL* = −694.13, AIC = −680.13). The test of fixed effects and the parameter estimates showed that the linear, B = 0.007, F_1,326.40_ = 7.45, *p* = 0.007, trend significantly described the pattern of the data over time. The low positive value of the linear effect indicates a tendency of the TEAC to increase over time. The estimation of covariance parameters showed that the variance of random intercepts was Var(u_0j_) = 0.003 assuming that intercepts value of TEAC varied significantly across young tennis player (*p* = 0.000). The variance of the tennis players’ slopes Var(u_1j_) = 0.00 and covariance between intercepts and slopes (01.00) showed redundant covariance parameters.

### 3.5. sAA

The lowest average value of sAA was found in the baseline, while the highest values were found in the pre-competition and post-match shot, showing an irregular distribution (Figure 5a). Figure 5b shows that the mean trajectory of sAA indicated a general tendency to increase over the time but its slope gradually decreased. The assessment in chi-square change was significant (χ^2^ (1) = 10.37, *p* < 0.01) between the linear (−2*LL* = 4010.49, AIC = 4014.12) and quadratic polynomial (−2*LL* = 4000.12, AIC = 4013.94), but this was not significate (χ^2^ (1) = 2.17, *p* > 0.05) between the quadratic polynomial and cubic polynomial (−2*LL* = 3997.94, AIC = 1130.28). The test of fixed effects and the parameters estimates showed that the linear, B = 49.024, F_1,342.03_ = 20.19, *p* = 0.000, and quadratic, B = −5.164, F_1,338.14_ = 11.35, *p* = 0.001, trends both significantly described the pattern of the data over time. The positive linear effect indicated a general tendency of the sAA to increase in time, and the negative quadratic effect indicated that the trend of sAA tends to flatten out over time. The estimation of covariance parameters showed that the variance of random intercepts was Var(u_0j_) = 4697.90 assuming that baseline sAA varied significantly across young tennis player (*p* = 0.008). The variance of the tennis players’ slopes Var(u_1j_) = 13.73 and covariance between intercepts and slopes (0.89) showed redundant covariance parameters.

## 4. Discussion

The purpose of this paper was to describe the changes in biomarkers of oxidative and psychological stress in a group of children during a tennis tournament. This competition lasted one day and consisted of four matches that lasted from 16 to 23 min on average, having subjective effort intensities from 11 to 13 on the 6–20 scale [31]. Therefore, the participants carried out on four occasions during the same day a type of intermittent effort of average duration (more than 10 min and less than 30) and with RPE located in light and somewhat hard. These matches were not considered exhausting for the children based on previous studies [12].

Total antioxidant capacity (TAC) was evaluated by three different assays, namely TEAC, FRAS and CUPRAC. These assays reflect the activity of different antioxidant compounds and can offer a wide information about the antioxidant status of the individual [19]. In addition, uric acid that is considered as the major contributor of the TAC of the saliva, was measured. All the four biomarkers of antioxidant status showed a similar dynamic in the experimental trial of our study. They had lower values at baseline and showed a progressive increase during the four matches. This would in line with the increases in acid uric and other antioxidants reported in saliva after an acute exercise [18,36]. Increases in antioxidants in saliva could be to compensate the release of oxidants compounds that can appear after exercise, and in the particular case of acid uric, due to the increased purine oxidation and subsequent acid uric production [37]. Some studies in children and adolescents indicate an increase in antioxidants immediately after acute exercise such as GHS, catalase and TAC [14,38,39] and also increases in the antioxidant response in general after the practice of different sports (swimming, running, football and athletics) has been described [12].

No studies about the antioxidant response in children associated to the practice of tennis have been found. However, considering that the antioxidant capacity after physical effort in children depends mainly on the type of physical activity (intensity and duration) [14,38,40], we can postulate that the response to tennis would be similar to the response to other acute open exercises such, football and basketball. Basically, an acute open exercise requires the child to “play” a greater continuous cognitive interaction (perception-action) that in turn allows him/her to self-regulate its intensity, in contrast to acute closed exercise such as swimming of cycling that requires greater physical and cardiorespiratory demand with limited cognitive involvement (continuous movements). In this line it was described that the oxidative responses in children with the practice of football and basketball are similar [40] and similar increases in TAC to those found in our study were described after one day of playing basketball in children [41].

Interestingly, there was an increase in antioxidant values in the pre-competition sample in comparison with baseline. This also occurred with α-amylase and could be related with an anticipatory stress that has been previously described in other situations in which people are facing a stressful situation [42], caused mainly by the beginning of the competition in tennis [43]. These results are in agreement with other studies carried out on children who showed a significant increase in sAA just before karate and inline skating competitions [23,44]. This would indicate that the fact of facing the competition produces a psychological stress and increases sympathetic activity leading to increases in sAA. It could be postulated that this stress could also induce an increase in antioxidant biomarkers, as previously described [45]. The lack of changes in alfa-amylase due to exercise could reflect a particularity of the children or the fact that an acute but habitual exercise can not affect the adrenergic response [46].

This manuscript has several limitations. One was the lack of control group that would not make any exercise and therefore it could have evaluated the influence of circadian rhythms. According to previous reports, this would not have influence on TEAC but could have influenced the FRAS values [47]. Also results were not normalized by salivary flow or protein concentrations, although in case of some analytes in saliva such as uric acid or alpha-amylase this was not recommended in previous studies [18,48]. In addition, other factors such as the influence of pH changes on saliva analytes were not analyzed and this could be considered as a limitation of the manuscript as it may influence the values, as reported by other authors [49,50].

Overall, this should be considered as a pilot study in which the pattern of changes of salivary antioxidant biomarkers and sAA during a 1-day tennis competition has been established. This study can open new lines of investigation. For example, further research should be performed in order to elucidate if different competition systems could produce changes in the response of antioxidant biomarkers. Also, it would be of interest to evaluate if the ingestion of antioxidants previously to the competition could have any effect in the response of salivary biomarkers. In addition, it would be recommended to evaluate if the antioxidant response to this type of competition would depend of the age of the children as described in other sports [51] and if the biomarkers evaluated in our study could be related with the performance or the risk of lesion in the children.

## 5. Conclusions

One-day tennis competition in children showed a tendency to increase antioxidant biomarkers in saliva. In addition, there was an increase in pre-competition sAA possibly associated with psychological stress. Further studies should be conducted to assess the possible applications of these biomarkers to evaluate performance or risk of injury in this discipline and also to help coaches to develop better training planning and manage the volume of competition for young players.

## Figures and Tables

**Figure 1 ijerph-17-06269-f001:**
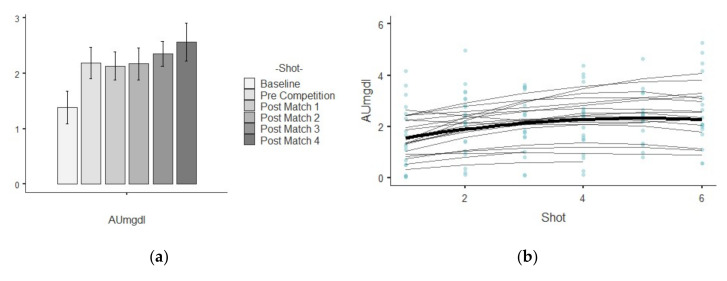
Illustration shows two plots related to AU parameters: (**a**) mean values and standard deviation in each shot temporarily ordered; (**b**) curvature of the mean trajectory over time (solid line) and the individual trajectories (thin lines) [quadratic effect], and the individual observed scores.

**Figure 2 ijerph-17-06269-f002:**
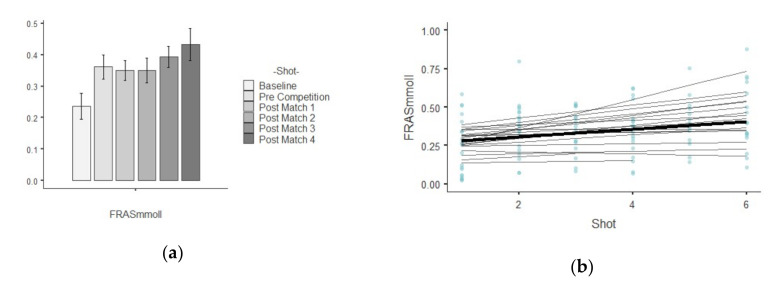
Illustration shows two plots related to FRAS parameters: (**a**) mean values and standard deviation in each shot temporarily ordered; (**b**) curvature of the mean trajectory over time (solid line) and the individual trajectories (thin lines) [linear effect], and the individual observed scores.

**Figure 3 ijerph-17-06269-f003:**
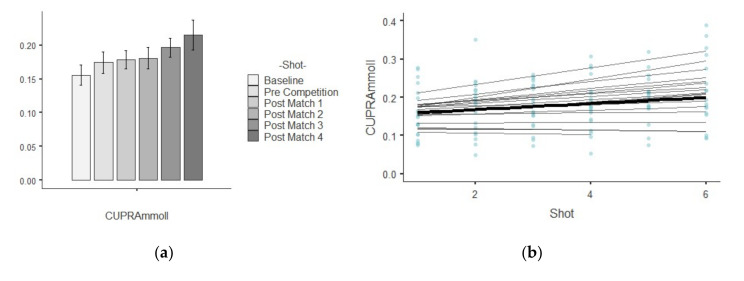
Illustration shows two plots related to CUPRA parameters: (**a**) mean values and standard deviation in each shot temporarily ordered; (**b**) curvature of the mean trajectory over time (solid line) and the individual trajectories (thin lines) [linear effect], and the individual observed scores.

**Figure 4 ijerph-17-06269-f004:**
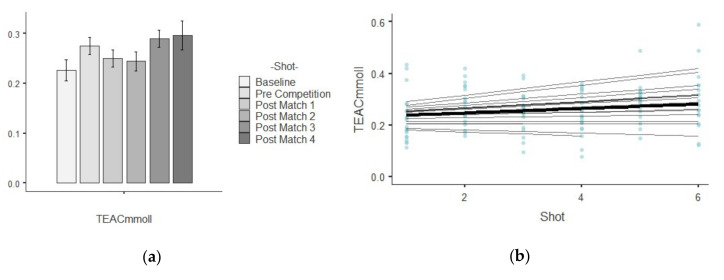
Illustration shows two plots related to TEAC parameters: (**a**) mean values and standard deviation in each shot temporarily ordered; (**b**) curvature of the mean trajectory over time (solid line) and the individual trajectories (thin lines) [linear effect], and the individual observed scores.

**Figure 5 ijerph-17-06269-f005:**
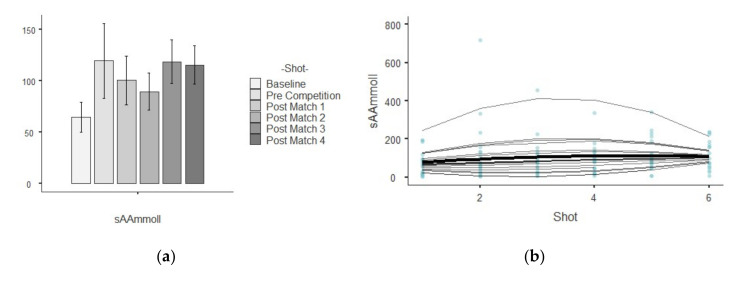
Illustration shows two plots related to sAA parameters: (**a**) mean values and standard deviation in each shot temporarily ordered; (**b**) curvature of the mean trajectory over time (solid line) and the individual trajectories (thin lines) [quadratic effect], and the individual observed scores.

**Table 1 ijerph-17-06269-t001:** Descriptive values (mean, standard deviation, variance, coefficient of variation, skewness and kurtosis) for each saliva parameters over time.

Saliva Parameter	Shot	Mean ± SD	Variance	CV	Skewness	Kurtosis
AU (μmol/L)	Baseline	1.38 ± 1.32	1.74	0.96	0.71	0.51
	Pre Competition	2.19 ± 1.28	1.63	0.58	0.06	0.51
	Post-match 1	2.14 ± 1.13	1.28	0.53	−0.31	0.52
	Post-match 2	2.17 ± 1.24	1.55	0.57	0.17	0.52
	Post-match 3	2.36 ± 0.95	0.90	0.40	0.46	0.54
	Post-match 4	2.57 ± 1.41	1.99	0.55	0.55	0.55
FRAS (mM/L)	Baseline	0.23 ± 0.18	0.03	0.81	0.44	−1.17
	Pre Competition	0.36 ± 0.17	0.03	0.48	0.37	1.17
	Post-match 1	0.35 ± 0.13	0.02	0.40	−0.45	−0.70
	Post-match 2	0.35 ± 0.16	0.03	0.48	0.02	−0.62
	Post-match 3	0.39 ± 0.14	0.02	0.36	0.62	1.78
	Post-match 4	0.43 ± 0.21	0.04	0.49	0.49	−0.38
CUPRA (mM/L)	Baseline	0.15 ± 0.06	0.00	0.40	0.63	−0.74
	Pre Competition	0.17 ± 0.07	0.01	0.41	0.36	0.60
	Post-match 1	0.17 ± 0.05	0.00	0.35	−0.23	−1.08
	Post-match 2	0.18 ± 0.07	0.00	0.38	0.20	−0.32
	Post-match 3	0.19 ± 0.06	0.00	0.32	−0.16	0.55
	Post-match 4	0.21 ± 0.09	0.01	0.43	0.48	−0.63
TEAC (mM/L))	Baseline	0.22 ± 0.09	0.01	0.41	1.05	0.40
	Pre Competition	0.27 ± 0.07	0.01	0.27	0.26	−0.85
	Post-match 1	0.25 ± 0.07	0.01	0.31	−0.16	0.12
	Post-match 2	0.24 ± 0.08	0.01	0.34	−0.36	−0.51
	Post-match 3	0.28 ± 0.07	0.01	0.25	0.61	2.72
	Post-match 4	0.29 ± 0.11	0.01	0.40	0.87	1.22
sAA (U/L)	Baseline	64.70 ± 64.50	4159.00	1.00	1.06	0.01
	Pre Competition	120.00 ± 163.00	26,636.00	1.36	2.94	9.98
	Post-match 1	101.00 ± 102.00	10,503.00	1.01	2.55	8.18
	Post-match 2	89.50 ± 78.30	6133.00	0.87	1.82	4.78
	Post-match 3	119.00 ± 89.90	8074.00	0.76	1.04	0.65
	Post-match 4	116.00 ± 76.70	5880.00	0.66	0.31	−1.31

AU = Uric Acid; FRAS = Ferric Reducing Ability of Saliva; CUPRA = Cupric Reducing Antioxidant Capacity; TEAC = Trolox Equivalent Antioxidant Capacity; sAA = Salivary alpha amylase; CV = coefficient of variation; *SD* = standard deviation.

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
