# Peer review of "Changes in Markers of Oxidative Stress and α-Amylase in Saliva of Children Associated with a Tennis Competition"

_ijerph, 2020, doi:10.3390/ijerph17176269_

Round 1
Reviewer 1 Report
The purpose of this paper was to analyze the changes caused by a one-day tennis tournament in biomarkers of oxidative stress in children. The four biomarkers of antioxidant status showed a similar dynamic with lower values at baseline and a progressive increase during the four matches. Distress-mediated tissue oxidative stress has been reported. These children were subjected to psychologically stressful conditions by tennis tournament. What is the significant to find these biomarker for these children? Are you suggesting to give these children antioxidants?Author Response
Dear reviewer,
Thank you very much for the comments and suggestions proposed for the article “Changes in markers of oxidative stress and alpha-amylase in saliva of children associated with a tennis competition”.
First of all, thanks for the useful suggestions and the opportunity to improve this document.
We are pleased to provide this letter to explain, point by point, the details of the revisions and modifications made to the manuscript and our responses to the comments. Any review is clearly highlighted in red within the manuscript.
We want to confirm that an exhaustive review has been carried out in accordance with the proposed considerations. The changes made are explained after reviewers comments.
REV.- The four biomarkers of antioxidant status showed a similar dynamic with lower values at baseline and a progressive increase during the four matches. Distress-mediated tissue oxidative stress has been reported. These children were subjected to psychologically stressful conditions by tennis tournament. What is the significant to find these biomarker for these children? Are you suggesting to give these children antioxidants?
AUT.- Thank you for the suggestion. The main intention of this work was to find out the oxidative responses of a group of children during a one-day tennis competition. Considering, on the one hand, the age (9-10 years) as they report that it is necessary to take into account
(https://journals.lww.com/jcma/Fulltext/2019/08000/Evaluation_of_oxidative_stress_and_antioxidant.10.aspx); and on the other hand, the frequency with which young people face similar competitive efforts because of their adherence to sports activities such as tennis. The results seem to show a slight tendency to increase throughout the competitive day, which may indicate greater muscular (AU, CUPRA, FRAP) and somatic (sAA) stress due to the fatigue of the sum of the 4 matches.
As for the need to take antioxidants by young athletes, it is not considered necessary. Since it seems that with these ages it allows their cushioning system to be generating adaptability, and optimizing the antioxidant capacity for when they are adults. As https://journals.lww.com/jcma/Fulltext/2019/08000/Evaluation_of_oxidative_stress_and_antioxidant.10.aspx suggests) that the younger the age, the more oxidation stress level and increased antioxidant capacity.
In addition, we think it is an interesting topic and we has been included more information in discussion.
You can find in the attached file the article with the modifications suggested by the three reviewers in "red".
Thank you very much.

Reviewer 2 Report
This is a simple paper that investigated changes in some parameters of oxidative stress in children after a tennis competition. In an overview, the paper needs a wide English editing, it presents methodological limitations and the results and conclusions do not bring new information for science or open consistent perspectives. Follows are some comments and suggestions.
Abstract
- Include more information about the sample such as gender and body mass.
- FRAP or FRAS? (Page 1, ine 24)
- Improve results presentation, more details are required.
- The conclusion is generalist and does not reflect the results found.
- The last sentence must be deleted.
Introduction
- Whats mean “inactive oxygen species”? (Page 1, line 39).
- Please, clarify this sentence ....” having clarified that muscle exercise (Page 1, line 39-40) promotes oxidative stress in people”... Important: if you write there are “several studies” then you must quote the references of several studies.
- What adaptations of you mean? (Page 2, line 44)
- What´s mean “stronger oxidative stress” (Page 2, line 44)
- “oxidative stress is one of the parameters that most affect performance in elite tennis players”. Are you sure about it? these references do not support your affirmation (Page 2, line 50-51)
- Please, include a reference to this affirmation (Page 2, line 63-64)
- repeated word “procedure” (Page 2, line 64)
- In the introduction is no need to justify the method. Here you need to give a sense of the need for this study, show the relevance, the gaps in the literature about it, and the contribution to Science. (Page 2, line 69-82)
Materials and Methods
- It is not a longitudinal study (Page 2, line 92)
- I suggest including the average time (months or year) of training (Page 3, line 95)
- Please, include the inclusion and exclusion criteria.
- Please, clarify the following information and add it in the paper: How were the samples stored after collection? How long between sample collection and processing? How long and under what conditions were the samples stored until analysis? Details of baseline collection are required. If the collection was performed before breakfast, how did this happen? Was breakfast in the same place as the competitions?
- It is necessary to add minimum information on the analysis methods of each KIT, such as references, adaptations, specificity and reference values.
- Barranco et al., (Ref 30) did not analyze Ferric Reducing Ability of Plasma (FRAP) just the Ferric Reducing Ability of Saliva (FRAS). Please, detail this method. If you used FRAP you should have collected the plasma, because the FRAPs kit is not specific for saliva.
- It was not clear to me the role of the salivary alpha-amylase towards the objectives of this study. Please clarify it.
Results
- In the legends of the tables and figures write the meaning of each acronym.
- I suggest including as table 1 the characteristics of the samples.
- Please identify the statistical difference between the collection times.
- I suggest that each parameter used is in a separate table.
- I consider it unnecessary to present the plots as illustrative figures.
Discussion
- The discussion is confused and does not discuss the data as expected.
- It is necessary to explain some results, for example. A) Why did the cupric reducing antioxidant capacity increase dramatically after the third and fourth matches?
- Why did the salivary alpha-amylase increased by 100% in the pre-competition and decreased after the first and second match?
Conclusion
- The conclusion does not correspond with the results found.
Author Response
Dear reviewer,
Thank you very much for the comments and suggestions proposed for the article “Changes in markers of oxidative stress and alpha-amylase in saliva of children associated with a tennis competition”.
First of all, thanks for the useful suggestions and the opportunity to improve this document.
We are pleased to provide this letter to explain, point by point, the details of the revisions and modifications made to the manuscript and our responses to the comments. Any review is clearly highlighted in red within the manuscript.
We want to confirm that an exhaustive review has been carried out in accordance with the proposed considerations. The changes made are explained after reviewers comments.
REV.- Abstract
REV.- Include more information about the sample such as gender and body mass.
AUT.-Thank you for your comments. Included in red on the manuscript.
REV.- FRAP or FRAS? (Page 1, ine 24)
AUT.- It is true, FRAP. It has been modified.
REV.- The conclusion is generalist and does not reflect the results found.
AUT.- Thanks. It has been modified.
REV.- The last sentence must be deleted.
AUT.- Thanks. It has been modified.
REV.- Introduction
REV.- Whats mean “inactive oxygen species”? (Page 1, line 39).
AUT.- Thank you. The correct name is “reactive oxygen species” sorry for this mistake. It has been modified.
REV.- Please, clarify this sentence ....” having clarified that muscle exercise (Page 1, line 39-40) promotes oxidative stress in people”... Important: if you write there are “several studies” then you must quote the references of several studies.
AUT.- Thank you. Studies have been included
REV.- What adaptations of you mean? (Page 2, line 44)
AUT.- Thank you. We refer to the metabolic adaptations caused by intense and long-lasting exercise. These seem to indicate that the ROS protects the individual's homeostasis in situations of metabolic demand. In addition, we has been included more information in introduction
REV.- What´s mean “stronger oxidative stress” (Page 2, line 44)
AUT.- Thank you. An oxidative stress with greater presence, with high oxidative load. We deleted “stronger” word.
REV.- “oxidative stress is one of the parameters that most affect performance in elite tennis players”. Are you sure about it? these references do not support your affirmation (Page 2, line 50-51).
AUT.- Thank you. The wording has been modified, referring to elite athletes in general.
REV.- Please, include a reference to this affirmation (Page 2, line 63-64)
AUT.- Thank you. It has been included
REV.- repeated word “procedure” (Page 2, line 64)
AUT.- Thank you. It has been changed.
REV.- In the introduction is no need to justify the method. Here you need to give a sense of the need for this study, show the relevance, the gaps in the literature about it, and the contribution to Science. (Page 2, line 69-82)
We agreed with the reviewer that in general in the introduction there is no need to justify the method. However, in this particular situacion we wanted to stress two facts:
- That Trolox equivalent antioxidant capacity (TEAC), ferric reducing ability of plasma (FRAP), and cupric reducing antioxidant capacity (CUPRAC) are different assays described to determine TAC of a sample (19). Since in some cases in literature there is a confusion about the use and meaning of these assays.
- That uric acid is very abundant in saliva and therefore it is important to measure it.
However, if the reviewer thinks that we should delete this part or move it to another section of the manuscript we would be happy to do it.
REV.- Materials and Methods
REV.- It is not a longitudinal study (Page 2, line 92)
AUT.- Thank you. It has been changed.
REV.- I suggest including the average time (months or year) of training (Page 3, line 95)
AUT.- Thank you. It has been included.
REV.- Please, include the inclusion and exclusion criteria.
AUT. Thank you. It has been included
REV.- Please, clarify the following information and add it in the paper: How were the samples stored after collection? How long between sample collection and processing? How long and under what conditions were the samples stored until analysis? Details of baseline collection are required. If the collection was performed before breakfast, how did this happen? Was breakfast in the same place as the competitions?
AUT.- Thank you for your comment. All the samples were collected directly on the day of the competition, even the one before breakfast, as the players had breakfast at the competition complex. Immediately after their collection they were frozen at -80ºC until they were analysed.
REV.- It is necessary to add minimum information on the analysis methods of each KIT, such as references, adaptations, specificity and reference values. Barranco et al., (Ref 30) did not analyze Ferric Reducing Ability of Plasma (FRAP) just the Ferric Reducing Ability of Saliva (FRAS). Please, detail this method. If you used FRAP you should have collected the plasma, because the FRAPs kit is not specific for saliva.
AUT.- Thank you for your comments.
It is true, is FRAS; it has been changed. In addtion, we have been included a new reference (32) with more information.
REV.- It was not clear to me the role of the salivary alpha-amylase towards the objectives of this study. Please clarify it.
AUT.- Thank you for your comment. Alpha-amylase is very sensitive marker of psychological stress, and understanding that a competitive event was evaluated and that also the psychological stress could potentially influence the response of the oxidative biomarkers, we decided to include her to check her response
REV.- Results
REV.- In the legends of the tables and figures write the meaning of each acronym.
AUT.- The meaning of each acronym has been included in tables and figures.
REV.- I suggest including as table 1 the characteristics of the samples.
AUT.- Thank you. We consider that the characteristics of the sample appear in the method and maybe, is to repeat the same data twice.
REV.- Please identify the statistical difference between the collection times. I suggest that each parameter used is in a separate table. I consider it unnecessary to present the plots as illustrative figures.
AUT.- Thank you for your comments. We have considered including a baseline over time. This is why you have the results that way.
REV.- Discussion
REV.- The discussion is confused and does not discuss the data as expected.
AUT.- Thank you. We have tried to comment on the results and relate them to different studies.
REV.- It is necessary to explain some results, for example. A) Why did the cupric reducing antioxidant capacity increase dramatically after the third and fourth matches?
AUT.- Thank you for your comments. The table 1 has been modificated. We think that the results are better now.
REV.- Why did the salivary alpha-amylase increased by 100% in the pre-competition and decreased after the first and second match?
AUT.- Thank you. The beginning of the competition (pre-competition) the boys were feeling more nervous. This state of psychological stress was lost throughout the rest of the game, where the nerves were calming down. We have included it in the discussion.
REV.- Conclusion
REV.- The conclusion does not correspond with the results found.
AUT.- Thank you. It has been changed.
You can find in the attached file the article with the modifications suggested by the three reviewers in "red".
Thank you very much.

Reviewer 3 Report
Dear Authors, the manuscript describe the use of saliva in the field of physical activity. The idea is not a new concept, since several research groups proposed saliva analysis as a tool to monitor athlets performances and exercise in general. The new concept here, is the monitoring of oxidative stress in children during tennis game. Experimental plan is well designed and allow to confirm or not the initial hypothesis (i.e. monitoring OX using saliva). I have some generic comments:
- it is well know that collection procedures affects the chemical composition of saliva, in particular flow rate and pH may modify the uric acid concentration and alpha-amylase activity, respectively. (DOI: 10.1016/j.microc.2017.02.032 and https://doi.org/10.1016/j.microc.2017.04.033). In such cases, is extremely suggested to use the same collection procedure in order to compare data from different subjects and from different time-period. As example, the use of flow rate allows to calculate the amount of analytes exctreted per unit of time. According to these information, did you evaluate the flow rate and pH? Did you evalute the reproducibility of the collection procedure? Please discuss this point in the article in order to confirm or not the reliability of the proposed method.
- please, include all the analytical figures of merit of the proposed approached used to determine the target analytes in saliva.
- please, discuss the stability of the target analytes. Together with sampling collection, analytes stability is an additional variables that may impact the amount of analytes in real samples.
- please include the unit of measure in the table 1 and modify the digit numbers according to the variability of the methods.
- did you monitor the heart rate during exercise? alpha-amylase can be related to HR changes during exercise.
I suggest to clarify these points in the manuscript before to accept the paper in IJERPH.
Author Response
Dear reviewer,
Thank you very much for the comments and suggestions proposed for the article “Changes in markers of oxidative stress and alpha-amylase in saliva of children associated with a tennis competition”.
First of all, thanks for the useful suggestions and the opportunity to improve this document.
We are pleased to provide this letter to explain, point by point, the details of the revisions and modifications made to the manuscript and our responses to the comments. Any review is clearly highlighted in red within the manuscript.
We want to confirm that an exhaustive review has been carried out in accordance with the proposed considerations. The changes made are explained after reviewers comments.
Detailed explanation point by point:
REV.- it is well know that collection procedures affects the chemical composition of saliva, in particular flow rate and pH may modify the uric acid concentration and alpha-amylase activity, respectively. (DOI: 10.1016/j.microc.2017.02.032 and https://doi.org/10.1016/j.microc.2017.04.033). In such cases, is extremely suggested to use the same collection procedure in order to compare data from different subjects and from different time-period. As example, the use of flow rate allows to calculate the amount of analytes exctreted per unit of time. According to this information, did you evaluate the flow rate and pH? Did you evalute the reproducibility of the collection procedure? Please discuss this point in the article in order to confirm or not the reliability of the proposed method.
AUT.- Thank you for yor In a previous report we concluded that for UA measurements in saliva it would not be recommended to normalize the results by salivary flow or protein concentration, similarly we observed this with alpha-amylase (PLoS One 2017 Jun 27;12(6):e0180100. doi: 10.1371/journal.pone.0180100. eCollection 2017. Influence of the Way of Reporting alpha-Amylase Values in Saliva in Different Naturalistic Situations: A Pilot Study) so our collection method were reliable.
Homever, we have been included more information in discussion.
REV.- Please, discuss the stability of the target analytes. Together with sampling collection, analytes stability is an additional variables that may impact the amount of analytes in real samples.
AUT.- Thank you. All samples were stored at -80ºC and analysed before than 3 months. All analytes measured did not show significant changes at these conditions in a previous stability study (30).
REV.- Please include the unit of measure in the table 1 and modify the digit numbers according to the variability of the methods.
AUT.- Thank you. It has been included units of measurement following reference number 32
REV.- Did you monitor the heart rate during exercise? alpha-amylase can be related to HR changes during exercise.
AUT.- Thank you for your comments. Whether HR was monitored during the competition, extracting maximum, minimum and average values per game. And even though there is literature indicating the relationship between HR and alpha-amylase (https://journals.plos.org/plosone/article/file?id=10.1371/journal.pone.0127749&type=printable), let's not consider including cardiac data because the greatest increase in this biomarker was prior to the start of the first match.
You can find in the attached file the article with the modifications suggested by the three reviewers in "red".
Thank you very much.

Round 2
Reviewer 1 Report
Although there is an improvement on the revised version, grammar mistakes still exist.
Abstract:
1. don't need two decimal points for data such as year 9.46±0.66--to 9.5=/-0.7
2. The saliva variables analyzed were ---Biomarkers of antioxidant status measured from saliva were analzyed by
3. This report provides evidence that four competitive tennis matches a day cause a progressive increase in biomarkers of oxidative stress in children.
any solution for this finding? what is the application?
Introduction:
1. These seem to indicate that the ROS protects the individual's homeostasis in situations of metabolic demand.
Evidence? how? any antioxidative enzymes increased?
2. Therefore the main objective is to analyse the oxidative response in saliva during a tennis competition 87 in children, and compare it with the activity of alfa-amylase in saliva..
please clarify this ….
Methods:
- one decimal point is enough for data
- please put male ...
- A single measurement was made in all cases since all analytical methods showed an intra-assay imprecision lower than 15%, which indicated adequate assay repeatability, could you please put a (.)
Result
- Tbale 1 SAA (mmoll)--is this unit correct?
Discusion
- formate for pragaph 1 and 2 are different, please make it consistant.
- Some studies in children and adolescents indicate increased antioxidants immediately after acute exercise (14, 38, 39) also the practice of different sports (swimming, running, football and 283 athletics) effects in antioxidant capacity has been evaluated. Increase antioxidant? do you mean antioxidaive enzymes? what kind oxidant has been masured ?
- Considering, on the one hand, the age (9-10 years) as they report that it is necessary to take into account (47) and, on the other hand, the frequency with which young people face similar competitive efforts due to their attachment to sports activities such as tennis. Please clarify this sentence (on the other hand has been shown twice)
- The results seem to show a slight tendency to increase throughout the competitive day, a slight tendency is not very scitific , please alter
- As for the need to take antioxidants by young athletes, it is not considered necessary. Based on what evidence for you to make this conclusion, please clarify.
Author Response
Dear reviewer,
Thank you very much for the suggestions about our work, we have considered them in their entirety. The response letter in word is attached.
Greetings,
